# Childhood infection burden, recent antibiotic exposure and vascular phenotypes in preschool children

Angela Yu[1], Maria A. C. Jansen[2], Geertje W. Dalmeijer[2], Patricia Bruijning-Verhagen[2], Cornelis K. van der Ent[3], Diederick E. Grobbee[2], David P. Burgner[1,4,5‡]*, Cuno S. P. M. Uiterwaal[2‡]

1 Department of Paediatrics, Monash University, Clayton, Australia, 2 Julius Center for Health Sciences and Primary Care, University Medical Center Utrecht, Utrecht University, Utrecht, The Netherlands, 3 Department of Pediatric Pulmonology, Wilhelmina Children's Hospital, University Medical Center Utrecht, Utrecht University, Utrecht, The Netherlands, 4 Murdoch Children's Research Institute, Parkville, Australia, 5 Department of Paediatrics, Melbourne University, Parkville, Australia

☯ These authors contributed equally to this work.
‡ DPB and CSPMU also contributed equally to this work and consider as senior authors.
* david.burgner@mcri.edu.au

**Data Availability Statement:** There are some unavoidable restrictions regarding the availability of the data used in this study. The study was approved by the IRB a number of years ago and

## Abstract

### Background

Severe childhood infection has a dose-dependent association with adult cardiovascular events and with adverse cardiometabolic phenotypes. The relationship between cardiovascular outcomes and less severe childhood infections is unclear.

### Aim

To investigate the relationship between common, non-hospitalised infections, antibiotic exposure, and preclinical vascular phenotypes in young children.

### Design

A Dutch prospective population-derived birth cohort study.

### Methods

Participants were from the Wheezing-Illnesses-Study-Leidsche-Rijn (WHISTLER) birth cohort. We collected data from birth to 5 years on antibiotic prescriptions, general practitioner (GP)-diagnosed infections, and monthly parent-reported febrile illnesses (0–1 years). At 5 years, carotid intima-media thickness (CIMT), carotid artery distensibility, and blood pressure (BP) were measured. General linear regression models were adjusted for age, sex, smoke exposure, birth weight z-score, body mass index, and socioeconomic status.

### Results

Recent antibiotic exposure was associated with adverse cardiovascular phenotypes; each antibiotic prescription in the 3 and 6 months prior to vascular assessment was associated

there was no application to the IRB for data sharing at this time. If data sharing is requested, this would necessitate a review of this request by the responsible IRB. Queries regarding availability of the data used in this study should be addressed in the first instance to The Julius Center, Utrecht University Medical Center (email: SecretariaatJHN-3@umcutrecht.nl).

**Funding:** The WHISTLER birth cohort was supported with a grant from the Netherlands Organization for Health Research and Development (grant nr 2001-1-1322) and by an unrestricted grant from Glaxo Smith Kline Netherlands. WHISTLER-Cardio was supported with an unrestricted strategic grant from the University Medical Center Utrecht (UMCU), The Netherlands. DB is supported by an Investigator Grant (Leadership level 1; GTN1175744) from the National Health and Medical Research Council (Australia). Research at the Murdoch Children's Research Institute is supported by the Victorian Government's Operational Infrastructure Program. The funders had no role in study design, data collection and analysis, decision to publish, or preparation of the manuscript.

**Competing interests:** The authors have declared that no competing interests exist.

with an 18.1 μm (95% confidence interval, 4.5–31.6, $p$ = 0.01) and 10.7 μm (0.8–20.5, $p$ = 0.03) increase in CIMT, respectively. Each additional antibiotic prescription in the preceding 6 months was associated with an 8.3 mPa$^{-1}$ decrease in carotid distensibility (-15.6– -1.1, $p$ = 0.02). Any parent-reported febrile episode (compared to none) showed weak evidence of association with diastolic BP (1.6 mmHg increase, 0.04–3.1, $p$ = 0.04). GP-diagnosed infections were not associated with vascular phenotypes.

## Conclusions

Recent antibiotics are associated with adverse vascular phenotypes in early childhood. Mechanistic studies may differentiate antibiotic-related from infection-related effects and inform preventative strategies.

## Introduction

Cardiovascular disease (CVD) is the leading cause of adult morbidity and mortality worldwide [1, 2]. The underlying pathology in CVD is atherosclerosis, chronic inflammatory damage of the arterial wall that develops across the life course. Patients with CVD events increasingly present without traditional cardiovascular (CV) risk factors, such as dyslipidaemia, hypertension, and diabetes [3], and there is interest in novel determinants to inform prevention strategies earlier in life. Although clinical CVD occurs in adulthood, atherosclerosis begins in childhood [4], when infection burden is greatest. Infections are a ubiquitous and repeated cause of inflammation and have been associated with increased CVD risk and events in adults [5–7].

Cardiovascular risk factors in childhood predict adult CVD events in a dose-response manner [8]. Preclinical vascular phenotypes, including carotid intima-media thickness (CIMT) and carotid artery distensibility, are associated with traditional CV risk factors in both children and adults, and is a strong predictor of future CV events in adults [9–12].

There is a growing body of evidence linking severe infection burden in childhood and adult CVD risk and events [13–15], but the relationship between early life infection and preclinical vascular phenotypes in childhood is less clear [16, 17]. Antibiotic exposure, which is also common in childhood, has been associated with both CV risk factors, such as obesity, and with adverse vascular phenotypes in both children and adults, but findings are inconsistent [17–19].

We therefore aimed to investigate the longitudinal associations between childhood infection burden, antibiotic exposure and childhood preclinical vascular phenotypes in a prospective population-derived cohort.

## Methods

This study was part of the WHeezing-Illnesses-STudy-LEidsche-Rijn (WHISTLER) prospective birth cohort of 2996 healthy newborns that commenced in 2001 [20]. In 2007, the original study design was expanded to include measurements of vascular characteristics (The WHISTLER-Cardio cohort study). The study population live in a residential area near the city of Utrecht, the Netherlands, and include those from diverse social, cultural and socioeconomic backgrounds. Exclusion criteria are gestational age <36 weeks, major congenital abnormalities and neonatal respiratory disease. A flowchart of the study cohort is provided in Fig 1.

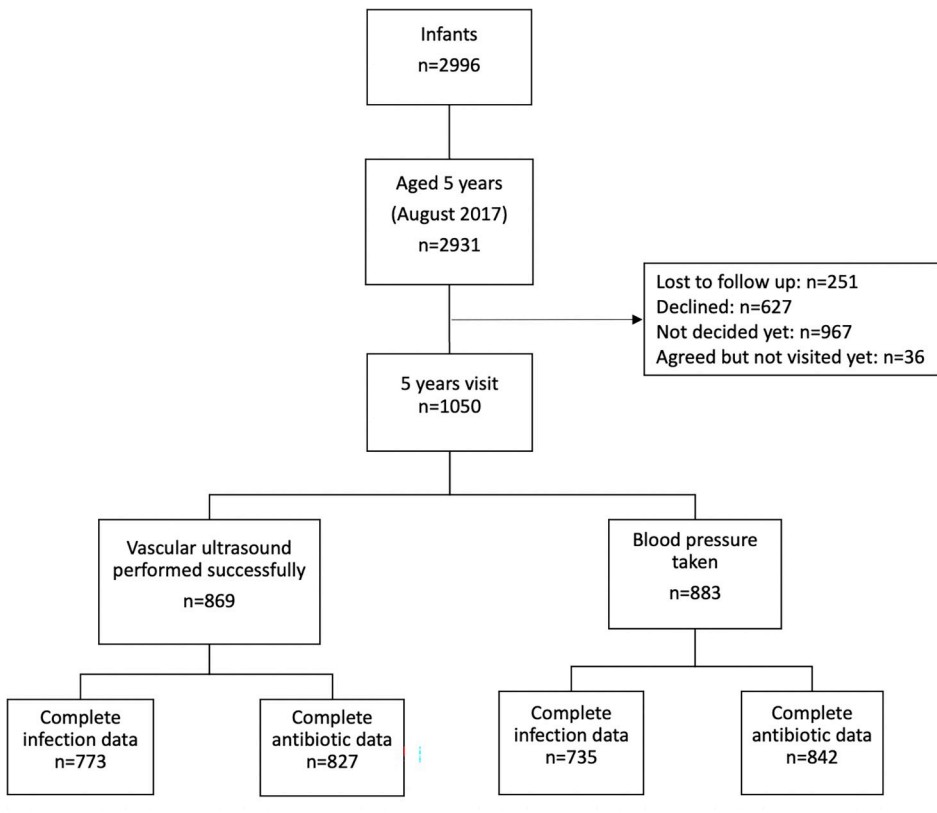

**Fig 1. Overview of the study population.**

The WHISTLER-Cardio cohort study was approved by the paediatric Medical Ethical Committee of the University Medical Center, Utrecht. Written informed parental consent was obtained.

## Neonatal visit

In the second to fourth week of life, families were invited to an ambulatory clinic where parents completed questionnaires regarding peri- and postnatal factors, including general characteristics, lifestyle factors and infections. Parental characteristics were obtained through questionnaires and from the linked Utrecht-Health-Project database, a large health-monitoring study [21]. Following the visit, parents were asked to complete a monthly health questionnaire for the child's first year of life, including number of days with fever over 38 degrees Celsius. Parental consent was sought to link the child's GP record.

## Childhood visits

Follow-up of children at age 5 years consisted of a parent questionnaire about early childhood lifestyle and health characteristics; number of infections in the preceding 12 months; and anthropometric, blood pressure (BP), carotid intima-media thickness (CIMT) and carotid distensibility measurements, as previously described [22]. Briefly, high-resolution echotracking technology (Art.Lab, Esaote, Italy), including a 128-radiofrequency line multiarray with a L10-5 40mm linear array transducer, was used to measure the far wall CIMT and distensibility (diastolic diameter and diastole to systole change in diameter) of the right common carotid artery.

Raw data were first analysed online and 6 second cineloops were stored without compression for offline analysis. Subjects were assessed in a supine position after at least 10 minutes rest. Each measurement (CIMT and distension) was repeated a maximum of four times and was carried out by five investigators and research nurses who were all blinded to other child characteristics. CIMT and diameter were measured at 2.1-μm resolution, and distension was measured with 1.7-μm resolution [23]. Mean coefficients of variation based on repeated, intra-individual measurements per child for CIMT, distension and lumen diastolic diameter were 5.3%, 5.9% and 2.3% respectively.

Blood pressure was measured three times on the right arm using an automatic device (DINA-MAP 8101-H6512, Model no 8101; Critikon, Tampa, Florida, USA). The first measurement was taken after 5 minutes rest, with 2 minutes of rest between measurements. Appropriate cuff sizes relative to the diameter of the arm were used. Mean systolic and diastolic BP (SBP; DBP) measurements taken in a sitting position were used in this analysis. CIMT, carotid distensibility and BP were measured successfully in 851, 731 and 883 children, respectively.

## Confounders and mediators

Several child and maternal factors were considered *a priori* to be potential confounders. Increasing age is related to adverse vascular parameters [24]. While it is disputed when sex differences in CIMT and distensibility are apparent in healthy children [24, 25], sex was considered a possible confounder and previous studies on pre-school children have adjusted for sex [17, 22]. The fully adjusted model additionally contained potential confounders including maternal smoking during pregnancy, birth weight, household smoking, socioeconomic status, and body mass index (BMI). Birth weight was adjusted for gestational age using linear regression to create a birth weight z-score.

## Infection burden

As primary measure of childhood infection burden, we recorded the number of GP-diagnosed infections up to age 5 years. We used linked GP International Classification of Primary Care (ICPC) diagnostic codes [26, 27] to identify infections that were likely to result in fever. The ICPC codes indicative of infection (S1 Appendix) were chosen *a priori* by infectious disease specialists (PBV and DPB). The infectious disease subgroup of ICPC component seven and previous studies related to GP-diagnosed infections were used to guide coding choice [17]. A period of at least 28 days between two GP-diagnosed infections was used to capture discrete infection episodes. As secondary measures of childhood infection burden, we included recent GP-diagnosed infections in the 3 and 6 months preceding the 5-year follow-up, and GP-diagnosed (ICPC code A03) and parent-reported (monthly diary) febrile episodes in the first year of life.

To investigate the correlation between GP diagnoses and parent reports, we analysed the number of GP and parent-reported infection and febrile illness in the same time periods. Parent-reported number of infection episodes in the 12 months preceding the 5-year follow-up visit was based on a single questionnaire and was used only to validate GP-diagnosed episodes during this time frame.

## Antibiotic exposure

Antibiotic prescriptions were obtained from a linked pharmacy registry [17, 28], which used the World Health Organisation Anatomical Therapeutic Chemical system (antimicrobials coded as J01) [29]. Antibiotic prophylaxis for urinary tract infections and topical antibiotics

were excluded as they are not prescribed for acute infection. The primary antibiotic exposure measure was the cumulative lifetime number of prescriptions from birth to age 5 years. In addition, we analysed recent antibiotic prescriptions in the 3, 6 and 12 months preceding vascular measurements.

### Data analysis

For a cumulative GP-diagnosed infection burden score, the number of infection codes per child from birth until the 5-year-old visit was summed. We calculated scores for recent GP-diagnosed infections; GP-diagnosed and parent-reported febrile episodes in the first year of life; and lifetime and recent number of antibiotic prescriptions. Means and variance measures of child and parent characteristics were compared between groups of GP-diagnosed infections (none and tertiles of number of infections) and number of parent-reported febrile episodes in the first year of life (none or at least one). Differences were tested using analysis of variance (ANOVA) for continuous variables and Chi-squared or Fisher's exact test for categorical variables and assessed for possible confounding. Kendall's tau correlation coefficients were calculated for the associations between GP and parent-reported infections in the 12 months preceding the 5-year-old follow-up, and separately for GP and parent-reported febrile episodes in the first year of life.

General linear regression was performed with GP-diagnosed infections; febrile episodes in the first year of life; and antibiotic exposures as independent variables in separate models. CIMT, carotid distensibility, and BP were dependent variables in separate models. The GP-diagnosed infection score from birth to age 5 years was analysed in three ways: as a dichotomous variable, a continuous variable, and as a categorical variable to detect linear and non-linear associations.

Analyses were repeated for recent GP-diagnosed infections; GP and parent-reported number of febrile episodes in the first year of life; and lifetime and recent antibiotic exposures as dichotomous and continuous independent variables. The minimally adjusted regression model included age and sex, while the adjusted model included all the potential *a priori* confounders. Results are expressed as linear regression coefficients with 95% confidence intervals (95% CI) and p-values. The significance threshold was defined as p-values less than 0.05. All analyses were performed using SPSS version 21.0 for Windows (IBM Corp., Armonk, New York, USA).

### Results

The median (IQR) of GP-diagnosed infections from birth to age 5 years was 3 (1–6) and 92% of children (711/773) had at least one infection recorded. The median number of parent-reported febrile episodes in the first year of life was 5 (2–10) and 82% of children (634/775) had at least one febrile episode reported. The median number of antibiotic prescriptions from birth to age 5 years was 1 (0–3) and 63% (583/928) received at least one antibiotic prescription. Of the 842, antibiotics were prescribed to 4%, 7%, and 15% of children in the 3, 6, and 12 months preceding vascular measurements, respectively.

Cohort mean (SD) CIMT was 388.6 μm (42.3), mean carotid distensibility was 94.1 mPa$^{-1}$ (25.8), mean SBP was 104.9 mmHg (7.5), and mean DBP was 54.3 mmHg (7.3) at age 5 years. Tables 1 and 2 and S1 Table show baseline characteristics of the participating children and their parents. The first child in the family and those with more GP-diagnosed allergies had more infections (Table 1). Shorter exclusive breastfeeding duration and those with more GP-diagnosed allergy and parental allergy had more antibiotic prescriptions (Table 2). There were weak correlations between GP-diagnosed and parent-reported infections in the preceding 12

**Table 1. Baseline characteristics–general practitioner-diagnosed infections.**

| Characteristics | Number of general practitioner diagnosed childhood infectious diseases before age 5 years | | | | | |
| --- | --- | --- | --- | --- | --- | --- |
| | 0 | 1–2 | 3–5 | ≥6 | Total | p-value |
| | n = 67 | n = 199 | n = 259 | n = 248 | n = 773 | |
| Infancy (0–4 weeks of age) | | | | | | |
| Male (n, %) | 28 (41.8) | 88 (44.7) | 129 (50) | 127 (51.6) | 372 (48.4) | 0.31 |
| Vaginal and assisted mode of delivery (n, %) | 53 (82.8) | 169 (86.7) | 210 (84.0) | 195 (82.3) | 627 (84.0) | 0.55 |
| Gestational age (w) | 40.1 (1.4) | 40.0 (1.2) | 40.0 (1.4) | 39.9 (1.3) | 39.9 (1.3) | 0.84 |
| Birth weight (g) | 3614 (531) | 3505 (537) | 3601 (490) | 3527 (471) | 3555 (501) | 0.13 |
| Birth weight (z-score) | 0.10 (1.0) | -0.10 (1.0) | 0.12 (1.0) | -0.03 (1.0) | 0.02 (1.0) | 0.12 |
| Maternal age at birth (yr) | 33.4 (3.5) | 32.9 (3.7) | 32.8 (3.3) | 32.7 (4.0) | 32.8 (3.7) | 0.57 |
| First child in family (n, %) | 26 (40.0) | 81 (44.0) | 112 (45.2) | 126 (53.8) | 345 (47.2) | 0.08 |
| Smoke exposure in pregnancy (yes) (n, %) | 9 (13.8) | 35 (18.7) | 48 (19.0) | 43 (18.3) | 135 (18.2) | 0.81 |
| Exclusive breastfeeding duration (days) | 91.3 (109.2) | 72.7 (74.7) | 81.6 (84.4) | 70.1 (84.2) | 76.6 (84.5) | 0.21 |
| European-caucasian[a] ethnicity child (n, %) | 49 (92.5) | 144 (94.7) | 187 (93.0) | 161 (86.1) | 541 (91.2) | 0.02 |
| Childhood (at pre-school visit) | | | | | | |
| Age (yr) | 5.4 (0.2) | 5.5 (0.3) | 5.5 (0.3) | 5.5 (0.3) | 5.5 (0.3) | 0.36 |
| BMI (kg/m$^2$) | 15.0 (1.5) | 15.1 (1.1) | 15.3 (1.6) | 15.3 (1.4) | 15.2 (1.4) | 0.16 |
| Household smoke exposure in childhood (yes) (n, %) | 4 (6.1) | 17 (8.7) | 12 (4.8) | 13 (5.4) | 46 (6.1) | 0.38 |
| GP-diagnosed allergy (n, %) | 31 (46.3) | 77 (38.7) | 119 (45.9) | 140 (56.5) | 367 (47.5) | 0.002 |
| Parental characteristics | | | | | | |
| Parent allergy (n, %) | 32 (52.5) | 100 (56.5) | 144 (61.0) | 139 (64.4) | 415 (60.1) | 0.25 |
| Grandparent with premature CVD (n, %) | 28 (41.8) | 60 (30.2) | 76 (29.3) | 82 (33.1) | 246 (31.8) | 0.24 |
| Parent with premature CVD (n, %) | 3 (4.5) | 3 (1.5) | 4 (1.5) | 12 (4.8) | 22 (2.8) | 0.06 |
| Mother's BMI (kg/m$^2$) | 24.8 (4.9) | 24.4 (3.6) | 25.2 (4.0) | 24.7 (4.4) | 24.8 (4.1) | 0.24 |
| Father's BMI (kg/m$^2$) | 25.6 (3.7) | 25.1 (2.8) | 25.0 (3.1) | 25.8 (3.2) | 25.3 (3.1) | 0.05 |
| Mothers with tertiary education (n, %) | 43 (76.8) | 116 (68.6) | 158 (70.2) | 127 (61.4) | 444 (67.6) | 0.09 |

[a]Child European-caucasian if both parents born in European-caucasian countries (according to Center for Statistics Netherlands

http://www.cbs.nl/nl-NL/menu/methoden/begrippen/default.htm?ConceptID=1057)

Values are mean (SD) unless otherwise indicated

months (Kendall's tau = 0.15, p<0.001) and febrile episodes in the first year of life (tau = 0.07, p = 0.03).

There was no association between GP-diagnosed infections and CIMT, carotid distensibility or BP at age 5 years (Tables 3 and 4). Recent GP-diagnosed infections in the preceding 3 and 6-months were not associated with vascular phenotypes (S2 Table). Neither GP-diagnosed nor parent-reported febrile episodes in the first year of life were associated with CIMT or carotid distensibility (Table 5). Table 6 shows weak evidence of the following associations with DBP: any versus no GP-diagnosed febrile illness in the first year of life (β: 1.9 mmHg, p = 0.07); each additional GP-diagnosed febrile illness (1.6 mmHg p = 0.05); any versus no parent-reported febrile episode (1.6 mmHg, p = 0.04); and any versus no parent-reported febrile episode and SBP (1.5 mmHg, p = 0.05). There was no association between the continuous number of parent-reported febrile episodes and SBP, nor GP-diagnosed febrile illness and SBP.

The cumulative lifetime number of antibiotic prescriptions was not associated with CIMT or carotid distensibility (Table 7). In adjusted models, each additional antibiotic prescription in the 3 and 6 months prior to vascular measurement was associated with an 18.1 μm (p = 0.01) and 10.7 μm (p = 0.03) increase in CIMT, respectively. Antibiotic prescriptions in

**Table 2. Baseline characteristics–lifetime antibiotic prescription(s).**

| Characteristics | Lifetime antibiotic prescription(s) | | Total | |
| --- | --- | --- | --- | --- |
| | None | One or more | | p-value |
| | n = 345 | n = 583 | n = 928 | |
| Infancy (0–4 weeks of age) | | | | |
| Male (n, %) | 147 (43.0) | 301 (52.0) | 448 (48.6) | 0.009 |
| Vaginal and assisted mode of delivery (n, %) | 291 (86.9) | 468 (83.4) | 759 (84.7) | 0.16 |
| Gestational age (w) | 40.0 (1.3) | 39.9 (1.3) | 39.9 (1.3) | 0.51 |
| Birth weight (g) | 3528 (499) | 3560 (498) | 3548 (498) | 0.36 |
| Birth weight (z-score) | -0.08 (1.0) | 0.02 (1.0) | -0.02 (1.0) | 0.16 |
| Maternal age at birth (yr) | 32.5 (5.3) | 32.4 (5.1) | 32.4 (5.2) | 0.63 |
| First child in family (n, %) | 143 (43.1) | 255 (46.4) | 398 (45.2) | 0.33 |
| Smoke exposure in pregnancy (n, %) | 57 (17.1) | 102 (18.3) | 159 (17.9) | 0.65 |
| Exclusive breastfeeding duration (days) | 88.4 (87.3) | 71.3 (80.8) | 77.7 (83.7) | 0.003 |
| European-caucasian[a] ethnicity child (n, %) | 260 (93.9) | 398 (90.7) | 658 (91.9) | 0.13 |
| Childhood at pre-school visit | | | | |
| Age (yr) | 5.4 (0.6) | 5.5 (0.3) | 5.4 (0.4) | 0.04 |
| BMI (kg/m²) | 15.1 (1.3) | 15.2 (1.4) | 15.2 (1.4) | 0.26 |
| Household smoke exposure in childhood (yes) (n, %) | 18 (5.3) | 32 (5.7) | 50 (5.5) | 0.81 |
| GP-diagnosed allergy (n, %) | 90 (26.1) | 279 (47.9) | 369 (39.8) | <0.001 |
| Parental characteristics | | | | |
| Parent allergy (n, %) | 163 (55.1) | 300 (62.3) | 463 (59.6) | 0.04 |
| Grandparent with premature CVD (n, %) | 115 (33.3) | 180 (30.9) | 295 (31.8) | 0.44 |
| Parent with premature CVD (n, %) | 10 (2.9) | 16 (2.7) | 26 (2.8) | 0.89 |
| Mother's BMI (kg/m²) | 24.9 (4.1) | 24.8 (4.1) | 24.8 (4.1) | 0.71 |
| Father's BMI (kg/m²) | 25.3 (3.2) | 25.3 (3.0) | 25.3 (3.1) | 0.87 |
| Mothers with tertiary education (n, %) | 203 (68.8) | 323 (66.1) | 526 (67.1) | 0.43 |

[a]Child European-caucasian if both parents born in European-caucasian countries (according to Center for Statistics Netherlands
http://www.cbs.nl/nl-NL/menu/methoden/begrippen/default.htm?ConceptID=1057)

Values are mean (SD) unless otherwise indicated

the 12 months preceding were not associated with CIMT (2.2 μm, $p = 0.47$). In the 6 months preceding vascular measurement, each additional antibiotic prescription was associated with an 8.3 mPa$^{-1}$ decrease in carotid distensibility ($p = 0.02$) and any antibiotic prescription was associated with a 10.9 mPa$^{-1}$ decrease in carotid distensibility compared to no prescription ($p = 0.02$). Antibiotic prescriptions in the 3 and 12 months preceding vascular measurement were not associated with carotid distensibility. Antibiotic prescriptions were not associated with BP at age 5 years (S3 Table).

## Discussion

This is the first large, population-based study to report modest but consistent evidence for an association between recent antibiotic use and adverse vascular phenotypes at 5 years of age. These associations were robust to adjustment for *a priori* determined confounders. More recent antibiotic use was associated with increased CIMT, although the absolute increase was modest and the long-term clinical implications unknown. We found no evidence of an association between overall or recent GP-diagnosed infections and vascular phenotypes. There was weak evidence that febrile illness in the first year of life may be associated with a small increase in DBP at 12 months of age.

**Table 3. Infections in first 5 years of life and carotid artery characteristics at age 5 years.**

| GP-diagnosed infections | Model | Carotid intima-media thickness (μm) | | | Carotid distensibility (mPa$^{-1}$) | | |
|---|---|---|---|---|---|---|---|
| | | N | Linear regression coefficient (95% CI) | p-value | N | Linear regression coefficient (95% CI) | p-value |
| Any vs none | Unadjusted | 695/760 | -3.7 (-14.5, 7.0) | 0.50 | 591/649 | 1.6 (-5.6, 8.8) | 0.66 |
| | Minimally adjusted* | 690/755 | -5.2 (-15.8, 5.5) | 0.34 | 588/646 | 2.1 (-5.0, 9.3) | 0.56 |
| | Adjustedł | 562/621 | -9.2 (-20.9, 2.4) | 0.12 | 482/532 | 2.2 (-5.9, 10.3) | 0.59 |
| Total number | Unadjusted | 695/760 | -0.2 (-0.6, 0.9) | 0.69 | 591/649 | 0.06 (-0.5, 0.7) | 0.82 |
| | Minimally adjusted* | 690/755 | 0.02 (-0.7, 0.8) | 0.97 | 588/646 | 0.1 (-0.4, 0.6) | 0.66 |
| | Adjustedł | 562/621 | -0.4 (-1.2, 0.5) | 0.42 | 482/532 | 0.2 (-0.4, 0.8) | 0.48 |
| Total number categories | | | | | | | |
| 0 | Unadjusted | 65/760 | ref | | 58/649 | ref | |
| 1 or 2 | | 196/760 | -3.8 (-15.6, 8.1) | 0.54 | 165/649 | 0.4 (-7.5, 8.4) | 0.91 |
| 3 to 5 | | 254/760 | -4.7 (-16.2, 6.9) | 0.43 | 218/649 | 1.6 (-6.2, 9.3) | 0.69 |
| 6 to 33 | | 245/760 | -2.8 (-14.3, 8.8) | 0.64 | 208/649 | 2.5 (-5.2, 10.3) | 0.52 |
| 0 | Minimally adjusted* | 65/755 | ref | | 58/646 | ref | |
| 1 or 2 | | 194/755 | -4.5 (-16.2, 7.3) | 0.46 | 163/646 | 0.7 (-7.2, 8.7) | 0.85 |
| 3 to 5 | | 253/755 | -6.1 (-17.5, 5.4) | 0.30 | 217/646 | 2.1 (-5.6, 9.8) | 0.60 |
| 6 to 33 | | 243/755 | -4.7 (-16.3, 6.8) | 0.42 | 208/646 | 3.3 (-4.5, 11.0) | 0.41 |
| 0 | Adjustedł | 54/621 | ref | | 47/532 | ref | |
| 1 or 2 | | 158/621 | -7.2 (-20.1, 5.7) | 0.27 | 132/532 | -0.3 (-9.3, 8.7) | 0.95 |
| 3 to 5 | | 212/621 | -9.9 (-22.4, 2.6) | 0.12 | 186/532 | 2.0 (-6.6, 10.6) | 0.65 |
| 6 to 33 | | 192/621 | -10.2 (-22.9, 2.4) | 0.11 | 164/532 | 4.6 (-4.2, 13.4) | 0.30 |

*Minimally adjusted: age and sex.

łAdjusted: age, sex, pregnancy and childhood household smoking, BMI, birth weight z-score, and SES.

The WHISTLER cohort allows for extensive confounder adjustment, decreasing the likelihood that the observed associations are due to residual confounding. There is minimal risk of information bias as investigators conducting vascular measurements were blinded to child characteristics. Moreover, measurement of arterial phenotypes is largely automated. As WHISTLER is a population-based cohort, health conscious families may be more willing to participate and complete follow-up. Study participants included in this analysis were not different from those excluded, apart from maternal smoking during pregnancy, which was higher in the group that completed follow-up. The antibiotic exposure data are more robust than self-reports as they were collected prospectively at the time of GP consultation and antibiotic dispensation. GP consultations are free of charge in the Netherlands, reducing confounding by socioeconomic status.

**Table 4. Infections in first 5 years of life and blood pressure at age 5 years.**

| GP-diagnosed infections | Model | N | Systolic blood pressure (mmHg) | | N | Diastolic blood pressure (mmHg) | |
|---|---|---|---|---|---|---|---|
| | | | Linear regression coefficient (95% CI) | p-value | | Linear regression coefficient (95% CI) | p-value |
| Any vs none | Unadjusted | 666/731 | -0.6 (-2.5, 1.3) | 0.55 | 666/731 | -0.8 (-2.6, 1.0) | 0.37 |
| | Minimally adjusted* | 661/726 | -0.6 (-2.5, 1.3) | 0.54 | 661/726 | -0.9 (-2.7, 0.9) | 0.32 |
| | Adjustedł | 537/590 | -1.4 (-3.5, 0.7) | 0.18 | 537/590 | -0.7 (-2.8, 1.3) | 0.49 |
| Total number | Unadjusted | 666/731 | -0.1 (-0.2, 0.1) | 0.47 | 666/731 | -0.03 (-0.2, 0.1) | 0.64 |
| | Minimally adjusted* | 661/726 | -0.1 (-0.2, 0.1) | 0.47 | 661/726 | -0.04 (-0.2, 0.1) | 0.57 |
| | Adjustedł | 537/590 | -0.1 (-0.3, 0.1) | 0.19 | 537/590 | -0.03 (-0.2, 0.1) | 0.73 |
| Total number categories | | | | | | | |
| 0 | Unadjusted | 65/731 | ref | | 65/731 | ref | |
| 1 or 2 | | 191/731 | 0.1 (-2.0, 2.2) | 0.93 | 191/731 | -0.5 (-2.5, 1.5) | 0.65 |
| 3 to 5 | | 242/731 | -0.7 (-2.8, 1.3) | 0.47 | 242/731 | -0.7 (-2.6, 1.3) | 0.49 |
| 6 to 33 | | 233/731 | -0.9 (-3.0, 1.1) | 0.36 | 233/731 | -1.3 (-3.2, 0.7) | 0.21 |
| 0 | Minimally adjusted* | 65/726 | ref | | 65/726 | ref | |
| 1 or 2 | | 189/726 | -0.006 (-2.1, 2.1) | 0.99 | 189/726 | -0.6 (-2.6, 1.4) | 0.58 |
| 3 to 5 | | 241/726 | -0.7 (-2.8, 1.3) | 0.47 | 241/726 | -0.7 (-2.7, 1.2) | 0.45 |
| 6 to 33 | | 231/726 | -0.9 (-3.0, 1.1) | 0.38 | 231/726 | -1.4 (-3.4, 0.6) | 0.16 |
| 0 | Adjustedł | 53/590 | ref | | 53/590 | ref | |
| 1 or 2 | | 153/590 | -0.9 (-3.2, 1.4) | 0.43 | 153/590 | -0.7 (-3.0, 1.5) | 0.52 |
| 3 to 5 | | 203/590 | -1.4 (-3.6, 0.8) | 0.21 | 203/590 | -0.6 (-2.8, 1.5) | 0.57 |
| 6 to 33 | | 181/590 | -1.9 (-4.2, 0.4) | 0.10 | 181/590 | -0.8 (-3.0, 1.4) | 0.47 |

*Minimally adjusted: age and sex.

łAdjusted: age, sex, pregnancy and childhood household smoking, BMI, birth weight z-score, and SES.

It proved extremely difficult to capture the burden of milder infections in childhood. GP diagnoses and parent reports of infection and febrile illness were poorly correlated. It is implausible that any 5-year-old child has no infections in their lifetime, though this was recorded for a small proportion of the children. Parents' health literacy and health-seeking behaviours are major confounders that are difficult to measure. GP diagnoses do not capture the complete infection burden in childhood, as the majority of infections do not result in a GP visit [30]. In our current study, infection remains a possible confounder in the association between antibiotic exposure and vascular phenotypes. Despite the large cohort, there was a limited number of children receiving antibiotics in the 3 and 6 months prior to vascular

**Table 5. Febrile episodes in the first year of life and carotid artery characteristics at age 5 years.**

| Febrile episodes | Model | Carotid intima-media thickness (μm) | | | Carotid distensibility (mPa$^{-1}$) | | |
|---|---|---|---|---|---|---|---|
| | | N | Linear regression coefficient (95% CI) | p-value | N | Linear regression coefficient (95% CI) | p-value |
| GP-diagnosed (any vs none) | Unadjusted | 69/760 | -2.3 (-12.7, 8.2) | 0.67 | 64/649 | 2.2 (-4.7, 9.0) | 0.54 |
| | Minimally adjusted* | 68/755 | -2.6 (-13.0, 7.9) | 0.63 | 64/646 | 2.4 (-4.4, 9.3) | 0.49 |
| | Adjustedł | 55/616 | -5.6 (-17.2, 5.9) | 0.34 | 52/529 | 2.6 (-5.1, 10.3) | 0.51 |
| GP-diagnosed (number) | Unadjusted | 69/760 | 0.8 (-7.1, 8.7) | 0.85 | 64/649 | 2.3 (-2.8, 7.5) | 0.38 |
| | Minimally adjusted* | 68/755 | 0.4 (-7.5, 8.3) | 0.92 | 64/646 | 2.6 (-2.6, 7.8) | 0.33 |
| | Adjustedł | 55/616 | -2.6 (-11.7, 6.5) | 0.57 | 52/529 | 3.2 (-2.8, 9.2) | 0.32 |
| Parent-reported (any vs none) | Unadjusted | 619/760 | 4.5 (-3.4, 12.3) | 0.26 | 530/651 | -0.1 (-5.3, 5.1) | 0.98 |
| | Minimally adjusted* | 614/753 | 3.4 (-4.4, 11.2) | 0.39 | 527/647 | 0.2 (-5.0, 5.4) | 0.94 |
| | Adjustedł | 546/656 | 2.3 (-6.4, 11.0) | 0.60 | 471/566 | 1.1 (-4.8, 7.0) | 0.71 |
| Parent-reported (number) | Unadjusted | 619/760 | 0.1 (-0.4, 0.5) | 0.71 | 530/651 | 0.2 (-0.1, 0.5) | 0.23 |
| | Minimally adjusted* | 614/753 | 0.04 (-0.4, 0.5) | 0.87 | 527/647 | 0.2 (-0.1, 0.5) | 0.23 |
| | Adjustedł | 546/656 | 0.1 (-0.4, 0.6) | 0.62 | 471/566 | 0.2 (-0.1, 0.6) | 0.16 |

*Minimally adjusted: age and sex.

łAdjusted: age, sex, pregnancy and childhood household smoking, BMI, birth weight z-score, and SES.

measurement, which resulted in some imprecision and limits the generalisability to other settings, where antibiotic prescribing in children is more profligate.

CIMT and distensibility are widely used in paediatric and adult studies of CV risk and are associated with traditional CV risk factors in both adults and children [9–11]. The significance of adverse vascular phenotypes in pre-school children is unclear, and it is unknown whether they persist into adulthood or if they are associated with CVD events.

The findings of the few previous studies investigating the relationship between antibiotic exposure and vascular phenotypes have been inconsistent. An earlier study on a smaller sub-population of the WHISTLER cohort found a similar association between antibiotics and adverse vascular phenotypes at age 5 years [17]. Conversely, a small prospective cohort study of children hospitalised with infection reported that antibiotics attenuated the increase in CIMT associated with infection, possibly due to reduced infection and/or inflammation severity or duration [18]. The study differs from ours in that the population had severe infection requiring hospitalisation that was associated with increased CIMT. Our study lacks data on childhood hospitalisations with infection, which are associated with CVD risk and events in older children and adults [13–15].

The most striking finding is the association of more recent antibiotic use with increased CIMT. This may reflect antibiotic use being a more robust marker of severe infection and associated inflammation, and/or an effect of antibiotics either directly, or mediated by changes to the microbiome. Antibiotic prescriptions may be a better indication of non-hospitalised, significant infections than parent reports or GP diagnoses. They are likely to capture a greater proportion of bacterial infections (particularly given the parsimonious use of antibiotics in the

**Table 6. Febrile episodes in the first year of life and blood pressure at age 5 years.**

| Febrile episodes | Model | Systolic blood pressure (mmHg) | | | Diastolic blood pressure (mmHg) | | |
|---|---|---|---|---|---|---|---|
| | | N | Linear regression coefficient (95% CI) | p-value | N | Linear regression coefficient (95% CI) | p-value |
| GP-diagnosed (any vs none) | Unadjusted | 69/731 | 0.8 (-1.1, 2.6) | 0.42 | 69/731 | 1.6 (-0.2, 3.3) | 0.08 |
| | Minimally adjusted* | 68/729 | 1.0 (-0.9, 2.8) | 0.30 | 68/729 | 1.5 (-0.2, 3.3) | 0.09 |
| | Adjustedł | 55/590 | 0.8 (-1.3, 2.8) | 0.46 | 55/590 | 1.9 (-0.1, 3.8) | 0.07 |
| GP-diagnosed (number) | Unadjusted | 69/731 | 0.2 (-1.2, 1.7) | 0.74 | 69/731 | 1.3 (-0.1, 2.7) | 0.06 |
| | Minimally adjusted* | 68/729 | 0.4 (-1.1, 1.8) | 0.62 | 68/729 | 1.3 (-0.1, 2.7) | 0.07 |
| | Adjustedł | 55/590 | 0.4 (-1.2, 2.1) | 0.61 | 55/590 | 1.6 (-0.01, 3.2) | 0.05 |
| Parent-reported (any vs none) | Unadjusted | 598/732 | 1.5 (0.1, 2.9) | 0.04 | 598/732 | 1.2 (-0.1, 2.6) | 0.07 |
| | Minimally adjusted* | 593/726 | 1.6 (0.2, 3.0) | 0.03 | 593/726 | 1.3 (-0.02, 2.7) | 0.05 |
| | Adjustedł | 525/630 | 1.5 (-0.01, 3.1) | 0.05 | 525/630 | 1.6 (0.04, 3.1) | 0.04 |
| Parent-reported (number) | Unadjusted | 598/732 | -0.01 (-0.1, 0.1) | 0.88 | 598/732 | 0.1 (-0.01, 0.1) | 0.10 |
| | Minimally adjusted* | 593/726 | 0.01 (-0.1, 0.1) | 0.86 | 593/726 | 0.1 (-0.01, 0.1) | 0.10 |
| | Adjustedł | 525/630 | 0.01 (-0.1, 0.1) | 0.90 | 525/630 | 0.1 (-0.01, 0.2) | 0.09 |

*Minimally adjusted: age and sex.

łAdjusted: age, sex, pregnancy and childhood household smoking, BMI, birth weight z-score, and SES.

Netherlands [31]), which generally result in more inflammation and may have more adverse effects on vasculature than viral infections, which generally result in a less marked inflammatory response [32].

Antibiotics affect the composition and function of gut microbiota (dysbiosis), changes that may persist for months to years, with potential long-term consequences on metabolism, inflammation, and the vasculature [33, 34]. The gut microbiota regulates the permeability of the intestinal barrier; dysbiosis may result in leakage of bacterial ligands, such as endotoxin (lipopolysaccharide, LPS), into the circulation, contributing to chronic inflammation [35, 36]. Western-diet-induced dysbiosis has been associated with increased circulating endotoxin and inflammatory markers, and increased atherosclerosis in a murine model [36]. Dysbiosis may also increase intestinal caloric uptake and affect the metabolism of lipids, leading to increased risk of overweight and obesity [37, 38].

Conversely, there is conflicting evidence about the metabolic changes resulting from antibiotic-induced dysbiosis. Antibiotics may moderate the atherogenicity of microbes, which has been shown in some mice models to reduce fat mass and lower inflammation [39]. In small human trials, antibiotic exposure was not associated with altered lipid profile or inflammation compared to placebo [40, 41]. Studies that address possible mechanisms underlying our epidemiological associations are warranted.

More recent antibiotic exposure was associated with more adverse vascular parameters. This suggests that the adverse effects of infection, inflammation, and/or dysbiosis are most significant in the short-term and may be transient. The adverse changes to children's vasculature from antibiotic exposure may become less evident with time, however their significance to long-term vascular health remains unknown.

**Table 7. Antibiotic prescriptions and carotid artery characteristics at age 5 years.**

| Antibiotic prescription | Model | N | Carotid intima-media thickness (μm) | | N | Carotid distensibility (mPa$^{-1}$) | |
|---|---|---|---|---|---|---|---|
| | | | Linear regression coefficient (95% CI) | p-value | | Linear regression coefficient (95% CI) | p-value |
| Lifetime (any vs none) | Unadjusted | 534/812 | 1.9 (-4.2, 8.1) | 0.54 | 458/696 | -2.3 (-6.4, 1.7) | 0.26 |
| | Minimally adjusted* | 520/790 | 0.8 (-5.5, 7.0) | 0.81 | 444/677 | -2.0 (-6.2, 2.1) | 0.34 |
| | Adjusted‡ | 431/666 | 0.3 (-6.5, 7.0) | 0.94 | 371/574 | -2.9 (-7.4, 1.7) | 0.21 |
| Lifetime (number) | Unadjusted | 534/812 | 0.4 (-1.1, 1.9) | 0.58 | 458/696 | -0.3 (-1.3, 0.7) | 0.55 |
| | Minimally adjusted* | 520/790 | 0.1 (-1.4, 1.7) | 0.85 | 444/677 | -0.3 (-1.3, 0.8) | 0.62 |
| | Adjusted‡ | 431/666 | 0.1 (-1.6, 1.7) | 0.95 | 371/574 | -0.4 (-1.6, 0.7) | 0.45 |
| Last 12 months (any vs none) | Unadjusted | 120/812 | 3.7 (-4.5, 11.9) | 0.38 | 102/696 | -2.7 (-8.1, 2.8) | 0.34 |
| | Minimally adjusted* | 119/790 | 3.6 (-4.6, 11.9) | 0.39 | 100/677 | -2.7 (-8.3, 2.8) | 0.33 |
| | Adjusted‡ | 96/666 | 0.4 (-8.8, 9.5) | 0.94 | 80/574 | -4.8 (-11.1, 1.4) | 0.13 |
| Last 12 months (number) | Unadjusted | 120/812 | 2.7 (-2.6, 8.0) | 0.31 | 102/696 | -0.6 (-4.2, 3.0) | 0.75 |
| | Minimally adjusted* | 119/790 | 3.0 (-2.4, 8.4) | 0.28 | 100/677 | -1.0 (-4.7, 2.6) | 0.58 |
| | Adjusted‡ | 96/666 | 2.2 (-3.7, 8.0) | 0.47 | 80/574 | -2.4 (-6.4, 1.7) | 0.25 |
| Last 6 months (any vs none) | Unadjusted | 57/812 | 7.8 (-3.6, 19.2) | 0.18 | 46/696 | -9.5 (-17.2, -1.7) | 0.02 |
| | Minimally adjusted* | 57/790 | 8.5 (-3.0, 19.9) | 0.15 | 45/677 | -9.2 (-17.1, -1.4) | 0.02 |
| | Adjusted‡ | 47/666 | 4.1 (-8.5, 16.6) | 0.53 | 37/574 | -10.9 (-19.7, -2.1) | 0.02 |
| Last 6 months (number) | Unadjusted | 57/812 | 9.9 (1.0, 18.7) | 0.03 | 46/696 | -5.9 (-12.2, 0.4) | 0.07 |
| | Minimally adjusted* | 57/790 | 10.3 (1.5, 19.1) | 0.02 | 45/677 | -5.7 (-12.1, 0.7) | 0.08 |
| | Adjusted‡ | 47/666 | 10.7 (0.8, 20.5) | 0.03 | 37/574 | -8.3 (-15.6, -1.1) | 0.02 |
| Last 3 months (any vs none) | Unadjusted | 30/812 | 11.8 (-3.7, 27.2) | 0.14 | 22/696 | -7.2 (-18.3, 3.8) | 0.20 |
| | Minimally adjusted* | 30/790 | 12.6 (-2.9, 28.0) | 0.11 | 21/677 | -6.7 (-18.1, 4.6) | 0.25 |
| | Adjusted‡ | 21/666 | 6.7 (-11.8, 25.2) | 0.48 | 13/574 | -11.6 (-26.2, 3.0) | 0.12 |
| Last 3 months (number) | Unadjusted | 30/812 | 17.1 (5.2, 29.0) | 0.01 | 22/696 | -6.0 (-14.6, 2.6) | 0.17 |
| | Minimally adjusted* | 30/790 | 17.5 (5.7, 29.4) | 0.004 | 21/677 | -5.6 (-14.4, 3.2) | 0.21 |
| | Adjusted‡ | 21/666 | 18.1 (4.5, 31.6) | 0.01 | 13/574 | -8.8 (-19.4, 1.9) | 0.11 |

*Minimally adjusted: age and sex.

‡Adjusted: age, sex, pregnancy and childhood household smoking, BMI, birth weight z-score, and SES.

There is evidence to suggest that total infection burden is associated with development of CVD [5–7, 16]. Previous studies on infection and CVD have been on retrospective serological evidence of infection [5–7], or on severe childhood infection episodes only [13–15]. Our study attempted to capture the total burden of childhood infections. However, we found little evidence of associations between both GP-diagnosed and parent-reported infections and vascular phenotypes. This is likely to reflect the inherent difficulties in objectively capturing the burden of less severe childhood infections due to factors such as parental experience and health

literacy that influence health-seeking behaviour, and possibly the substance of parent-reported data [30].

There was only weak evidence of an independent association between any GP-diagnosed and parent-reported febrile illness in the first year of life and a small increase in DBP, which was not seen with SBP. High childhood DBP is a risk factor for hypertension in adulthood [42]. The effect sizes of GP-diagnosed febrile episodes were larger than those of parent-reported, suggesting that GP-diagnosed febrile episodes are likely more severe. Febrile illness may be a better measure of significant infection than GP-diagnosed and parent-reported infections.

Future work in this area would include improved standardised measures of childhood infection and differentiating the effects of infection versus antibiotics on the vasculature. These could be aided through the use of biomarkers that reflect cumulative inflammatory insults, such as Glycoprotein acetyls (GlycA), despite the paucity of paediatric data [43]. Mechanistic studies, particularly related to the effect of different antibiotics on the microbiota, may inform future interventions [34]. Replication in other cohorts, especially those with different rates of antibiotic prescribing, would increase the validity of these data.

## Conclusions

We have shown an independent association between recent antibiotic exposure and adverse vascular phenotypes in healthy 5-year-old children. More recent antibiotic exposure was associated with increased effect size. Recent antibiotic exposure may have adverse effects on the childhood vasculature and mechanistic data would identify possible therapeutic interventions.

## Supporting information

**S1 Table. Baseline characteristics–parent-reported febrile days.**
(DOCX)

**S2 Table. Recent infections and carotid artery characteristics at age 5 years.**
(DOCX)

**S3 Table. Antibiotic prescriptions and blood pressure at age 5 years.**
(DOCX)

**S1 Appendix. International Classification of Primary Care (ICPC) codes included in general practitioner (GP) diagnosed infections.**
(DOCX)

## Acknowledgments

The authors would like to thank all the participating families and children.

## Author Contributions

**Conceptualization:** Cornelis K. van der Ent, Diederick E. Grobbee, David P. Burgner, Cuno S. P. M. Uiterwaal.

**Data curation:** Geertje W. Dalmeijer.

**Formal analysis:** Angela Yu, Maria A. C. Jansen, Cuno S. P. M. Uiterwaal.

**Methodology:** Angela Yu, Maria A. C. Jansen, Geertje W. Dalmeijer, Patricia Bruijning-Verhagen, David P. Burgner, Cuno S. P. M. Uiterwaal.

**Supervision:** David P. Burgner, Cuno S. P. M. Uiterwaal.

**Writing – original draft:** Angela Yu, Maria A. C. Jansen.

**Writing – review & editing:** Angela Yu, Maria A. C. Jansen, Geertje W. Dalmeijer, Patricia Bruijning-Verhagen, Cornelis K. van der Ent, Diederick E. Grobbee, David P. Burgner, Cuno S. P. M. Uiterwaal.

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
