## [Decision Letter · Decision Letter 0]

31 Aug 2022

PONE-D-22-01018Childhood infection burden, recent antibiotic exposure and vascular phenotypes in preschool childrenPLOS ONE

Dear Dr. Burgner,

Thank you for submitting your manuscript to PLOS ONE. After careful consideration, we feel that it has merit but does not fully meet PLOS ONE’s publication criteria as it currently stands. Therefore, we invite you to submit a revised version of the manuscript that addresses the points raised during the review process. Your manuscript has been evaluated by three reviewers, their comments are included below. The reviewers provided positive feedback about this submission, but they raised a few points that need to be addressed through minor revisions. Reviewer 2 asked for an additional analysis (table) to address the potentially confounding factor of allergy burden. The other comments could be addressed with minor text revisions. Please address all of the reviewers' comments through revisions to your manuscript and in your Response to Reviewers letter. Please submit your revised manuscript by Oct 14 2022 11:59PM. If you will need more time than this to complete your revisions, please reply to this message or contact the journal office at plosone@plos.org. Please include the following items when submitting your revised manuscript:A rebuttal letter that responds to each point raised by the academic editor and reviewer(s). You should upload this letter as a separate file labeled 'Response to Reviewers'.A marked-up copy of your manuscript that highlights changes made to the original version. You should upload this as a separate file labeled 'Revised Manuscript with Track Changes'.An unmarked version of your revised paper without tracked changes. You should upload this as a separate file labeled 'Manuscript'.If applicable, we recommend that you deposit your laboratory protocols in protocols.io to enhance the reproducibility of your results. Protocols.io assigns your protocol its own identifier (DOI) so that it can be cited independently in the future. For instructions see: https://journals.plos.org/plosone/s/submission-guidelines#loc-laboratory-protocols. Additionally, PLOS ONE offers an option for publishing peer-reviewed Lab Protocol articles, which describe protocols hosted on protocols.io. Read more information on sharing protocols at https://plos.org/protocols?utm_medium=editorial-email&utm_source=authorletters&utm_campaign=protocols.

We look forward to receiving your revised manuscript.

Kind regards,

Renee Hoch, Ph.D.

Managing Editor, PLOS Publication Ethics

PLOS ONE

Journal Requirements:

"The authors would like to thank all the participating families and children, and the WHISTLER study group."

"The WHISTLER birth cohort was supported with a grant from the Netherlands Organization for Health Research and Development (grant nr 2001-1-1322) and by an unrestricted grant from Glaxo Smith Kline Netherlands. WHISTLER-Cardio was supported with an unrestricted strategic grant from the University Medical Center Utrecht (UMCU), The Netherlands. DB is supported by an Investigator Grant (Leadership level 1; GTN1175744) from the National Health and Medical Research Council (Australia). Research at the Murdoch Children's Research Institute is supported by the Victorian Government's Operational Infrastructure Program. The funders had no role in study design, data collection and analysis, decision to publish, or preparation of the manuscript."

Reviewers' comments:

Reviewer's Responses to Questions

**Comments to the Author**

1. Is the manuscript technically sound, and do the data support the conclusions?

Reviewer #1: Yes

Reviewer #2: Partly

Reviewer #3: Yes

2. Has the statistical analysis been performed appropriately and rigorously? 

Reviewer #1: Yes

Reviewer #2: Yes

Reviewer #3: Yes

3. Have the authors made all data underlying the findings in their manuscript fully available?

Reviewer #1: Yes

Reviewer #2: Yes

Reviewer #3: Yes

4. Is the manuscript presented in an intelligible fashion and written in standard English?

Reviewer #1: Yes

Reviewer #2: Yes

Reviewer #3: Yes

5. Review Comments to the Author

Reviewer #1: Dear PLOS,

thank you for inviting me to review the submission entitled 'Childhood infection burden, recent antibiotic exposure and vascular phenotypes in preschool children'.

I am aware of the background research that these authors have done in this fascinating field of childhood factors influencing longer-term cardiovascular morbidity indicators.

In this regard, the authors are using a surrogate marker of CV disease - carotid intima-media thickness.

The hypothesis is intriguing - that the burden of common (inflammatory) childhood infection correlates with a long-term CV disease risk in this current era.

It is reasonable to infer burden of childhood infection being translated into antibiotic prescription by GP's to children.

The patient cohort is an invaluable reference (WHISTLER).

Exclusion criteria are also common-sense.

The infection burden criteria also were reasonable and informed by ID experts (PBV, DPB).

Statistical methodology appears appropriate.

The baseline demographics seem relatively broadly applicable to 'Western' countries - older maternal age, low percentage of smokers, European-caucasian ethnicity, a relatively highly educated cohort

(very high percentage of babies breast-fed exclusively).

minor comment:

Why would the first child in the family have more GP-diagnosed infections (slightly counter-intuitive unless this reflects anxious parents with their firstborn).

The results are well laid out and clear to follow.

The modest findings of recent antibiotic prescribing correlating weakly with changes in CIMT values is acknowledged and forms the basis of the discussion (with appropriate reference made to limitations / confounders).

the results aren't over-stated.

The abstract reflects the full manuscript accurately.

I think this study is of interest to the field of paediatrics and CV health based on this unique dataset for hypothesis testing and for the slight trend shown that will, no doubt, inform future studies by these authors and others.

Reviewer #2: I am very grateful for the opportunity to review this manuscript.

The article investigates the association between antibiotic use, common (non-sever) infections, and certain vascular phenotypes in young children.

The manuscript is well written and a great effort was done in the analysis. Data are presented in a detailed and intelligent fashion. The authors thoroughly discussed their findings and mentioned possible limitations of the study.

The important confounding factor that was not discussed enough by the authors is the allergy burden in the study population. The authors found that children with GP-diagnosed allergies had more infections (both GP-diagnosed infections and parent-reported febrile days). It was not mentioned though if they had more antibiotics prescriptions or not. I suggest adding a table to compare baseline characteristics between children classified according to antibiotic use. The importance of this table originated from the fact that the main positive study findings are more about recent antibiotic use rather than GP-diagnosed infections or family-reported infections.

The association between allergy especially allergic asthma and carotid intima-media thickness in children and adolescents has been suggested by different studies. This includes the study published in 2020 by some of the authors of this study and on the same (WHISTLER) cohort. They found that children with parental allergy with or without a GP-diagnosed allergy had a significantly higher CIMT compared to those not having such a history and diagnosis. While children with only a GP-diagnosed allergy had no difference in CIMT. [1] I think it is important for the authors to mention the data about parents’ allergies in this study as well (if possible) and include both parents' and children’s allergic history in the list of potential confounders before drawing the conclusion.

1. Annemieke MV Evelein, Frank LJ Visseren, Cornelis K van der Ent, Diederick E Grobbee, Cuno SPM Uiterwaal, Allergies are associated with arterial changes in young children, European Journal of Preventive Cardiology, Volume 22, Issue 11, 1 November 2015, Pages 1480–1487, https://doi.org/10.1177/2047487314554863

Reviewer #3: I would like to thank the authors and the editors for this opportunity to review this manuscript.

My comments are as below:

The second sentence about the background of the abstract was a bit confusing to me. Please edit it.

The introduction section:

1. Needs re-writing

2. Please present the facts with data from longitudinal studies, clinical trials, and systematic reviews.

3. Can you explain the long-term effects of childhood cardiovascular outcomes to rationalize your study better? Do these effects wane with age, and what does the existing literature state in this context?

Are there any prevalence estimates on childhood cardiovascular indices that result in cardiovascular diseases in adulthood?

In the exclusion criteria, please explain the reasons for not including children with neonatal respiratory diseases.

Sentence no 104: Please include the children's data collected from the GP record here.

Thank you.

6. PLOS authors have the option to publish the peer review history of their article (what does this mean?). If published, this will include your full peer review and any attached files.

Reviewer #1: **Yes: **Terence Prendiville

Reviewer #2: **Yes: **Mohammed Abdellatif

Reviewer #3: **Yes: **Sumanta Saha

---

## [Author Response · Author response to Decision Letter 0]

15 Feb 2023

We have verified that the grant numbers for awards received for our study match and are correct.

"The authors would like to thank all the participating families and children, and the WHISTLER study group."

"The WHISTLER birth cohort was supported with a grant from the Netherlands Organization for Health Research and Development (grant nr 2001-1-1322) and by an unrestricted grant from Glaxo Smith Kline Netherlands. WHISTLER-Cardio was supported with an unrestricted strategic grant from the University Medical Center Utrecht (UMCU), The Netherlands. DB is supported by an Investigator Grant (Leadership level 1; GTN1175744) from the National Health and Medical Research Council (Australia). Research at the Murdoch Children's Research Institute is supported by the Victorian Government's Operational Infrastructure Program. The funders had no role in study design, data collection and analysis, decision to publish, or preparation of the manuscript."

We have amended the Acknowledgements Section to remove any potential funding information, it now reads “The authors would like to thank all the participating families and children”. The Funding Statement as it currently reads is correct.

Data are available from the Julius Center for Health Sciences and Primary Care, University Medical Center Utrecht (contact via whistler@umcutrecht.nl) for researchers who meet the criteria for access to confidential data. All data are available for external parties with appropriate agreement on use and reference to the original investigators.

Reviewer #1:

Minor comment:

Why would the first child in the family have more GP-diagnosed infections (slightly counter-intuitive unless this reflects anxious parents with their firstborn).

We believe this does reflect the difference in first-time parents’ health literacy and health-seeking behaviours which we acknowledged in the discussion as likely confounders that are difficult to measure.

Reviewer #2:

The important confounding factor that was not discussed enough by the authors is the allergy burden in the study population. The authors found that children with GP-diagnosed allergies had more infections (both GP-diagnosed infections and parent-reported febrile days). It was not mentioned though if they had more antibiotics prescriptions or not. I suggest adding a table to compare baseline characteristics between children classified according to antibiotic use. The importance of this table originated from the fact that the main positive study findings are more about recent antibiotic use rather than GP-diagnosed infections or family-reported infections.

We appreciate the reviewer’s response to the manuscript. We have added the additional table of baseline characteristics of the participating children and their parents according to antibiotic prescription (Table 2, page 9, line 225).

The association between allergy especially allergic asthma and carotid intima-media thickness in children and adolescents has been suggested by different studies. This includes the study published in 2020 by some of the authors of this study and on the same (WHISTLER) cohort. They found that children with parental allergy with or without a GP-diagnosed allergy had a significantly higher CIMT compared to those not having such a history and diagnosis. While children with only a GP-diagnosed allergy had no difference in CIMT. [1] I think it is important for the authors to mention the data about parents’ allergies in this study as well (if possible) and include both parents' and children’s allergic history in the list of potential confounders before drawing the conclusion.

Thank you for your comments. We agree that the association between allergy and adverse vascular characteristics has previously been shown in children, including in the WHISTLER cohort. We have added parent allergy to both baseline characteristics tables (Table 1 and Table 2). Parent allergy is associated with increasing childhood antibiotic prescription and general practitioner diagnosed childhood infections, however the mechanisms underlying this relationship remain unclear and further investigations is outside of the scope of this study. We do not include allergy (childhood GP-diagnosed or parent allergy) as an a priori confounder as it is unlikely to directly affect the number of antibiotic prescriptions nor childhood infections. It is possible that allergies resulting in wheeze may be inappropriately treated with antibiotics, however this is likely uncommon in The Netherlands where there is parsimonious use of antibiotics and most GPs would interpret infection-associated wheeze as of viral aetiology and therefore would not prescribe antibiotics[29]. In addition, we believe that allergic wheeze lies on the causal pathway of antibiotic treatment to increased carotid intima-media thickness (CIMT). Allergic wheeze may lead to more inappropriate prescription of antibiotics, which in turn leads to increased CIMT. In this case, adjusting for allergies (including wheeze) in the analysis would filter out the effect of antibiotics on CIMT and would be methodologically incorrect.

Reviewer #3:

The second sentence about the background of the abstract was a bit confusing to me. Please edit it.

The second sentence about the background of the abstract was amended for clarity, it now reads:

“Severe childhood infection has a dose-dependent association with adult cardiovascular events and with adverse cardiometabolic phenotypes. The relationship between cardiovascular outcomes and less severe childhood infections is unclear.” (page 2, line 31)

The introduction section:

1. Needs re-writing

2. Please present the facts with data from longitudinal studies, clinical trials, and systematic reviews.

3. Can you explain the long-term effects of childhood cardiovascular outcomes to rationalize your study better? Do these effects wane with age, and what does the existing literature state in this context?

Thank you for your suggestions to improve the Introduction section. We have added the following clarification around the long-term effects off childhood cardiovascular outcomes in the Introduction:

“Cardiovascular risk factors in childhood predict adult CVD events in a dose-response manner.[8] Preclinical vascular phenotypes, including carotid intima-media thickness (CIMT) and carotid artery distensibility, are associated with traditional CV risk factors in both children and adults, and is a strong predictor of future CV events in adults.[9–12]” (Page 3, line 70-73)

We further discuss the implications of childhood cardiovascular outcomes in the Discussion section (see Page 15, line 331-334).

Are there any prevalence estimates on childhood cardiovascular indices that result in cardiovascular diseases in adulthood?

There are no longitudinal data that track these CV indices (by which we assume the reviewer means the carotid IMT and carotid distensibility) from preschool and primary school age children into CVD events in adulthood; cohorts have not been in existence long enough. Such studies are underway, but the participants, who were enrolled in pregnancy, are now only 10 years old. These key data will only be available to the next generation of researchers. We have shown that hospitalisation with infection in childhood is associated, in a dose-response manner, with CVD events in adulthood (PMID: 25938548), but these were population-level data and we do not have CIMT and other individual level indices.

In the exclusion criteria, please explain the reasons for not including children with neonatal respiratory diseases.

The WHISTLER study was initially conceptualized to investigate paediatric wheezing illness, and neonatal respiratory disease is a significant confounder in paediatric wheezing illnesses. This current study aimed to investigate non-severe, common illnesses in otherwise healthy pre-school children, therefore neonatal respiratory disease would preclude them from this population.

Sentence no 104: Please include the children's data collected from the GP record here.

We are not entirely clear what the reviewer is requesting. In terms of infection burden, we recorded the number of GP-diagnosed infections up to age 5 years using linked GP International Classification of Primary Care (ICPC) diagnostic codes. These data were collected and analysed after the Childhood visits at age 5 years occurred. The full method of GP data collection is stated in the section Infection Burden (Page 6, line 151-167).

---

## [Decision Letter · Decision Letter 1]

16 Jun 2023

PONE-D-22-01018R1Childhood infection burden, recent antibiotic exposure and vascular phenotypes in preschool childrenPLOS ONE

Dear Dr. Burgner,

Thank you for submitting your manuscript to PLOS ONE. After careful consideration, we feel that it has merit but does not fully meet PLOS ONE’s publication criteria as it currently stands. Therefore, we invite you to submit a revised version of the manuscript that addresses the points raised during the review process.

We look forward to receiving your revised manuscript.

Kind regards,

Engelbert Adamwaba Nonterah, MD, PhD

Academic Editor

PLOS ONE

Journal Requirements:

Reviewers' comments:

Reviewer's Responses to Questions

**Comments to the Author**

1. If the authors have adequately addressed your comments raised in a previous round of review and you feel that this manuscript is now acceptable for publication, you may indicate that here to bypass the “Comments to the Author” section, enter your conflict of interest statement in the “Confidential to Editor” section, and submit your "Accept" recommendation.

Reviewer #4: (No Response)

2. Is the manuscript technically sound, and do the data support the conclusions?

Reviewer #4: Yes

3. Has the statistical analysis been performed appropriately and rigorously? 

Reviewer #4: Yes

4. Have the authors made all data underlying the findings in their manuscript fully available?

Reviewer #4: Yes

5. Is the manuscript presented in an intelligible fashion and written in standard English?

Reviewer #4: Yes

6. Review Comments to the Author

Reviewer #4: Suggestion/s to the author:

The author should avoid repeating the numbers that are in tables (rather refer the reader to the specific table). If it is necessary, they can emphasize some of the key numbers for the reader that they think are most important. In the 1st paragraph of the results, I see that the author was showing the median and IQR, I suggest minor edits. Median and IQR are always together, therefore, no need to be continually repeating the "IQR". Maybe the author can write something like this “The median (IQR) of GP-diagnosed infections from birth to age 5 years was 3 (1-6) and 92% of children (711/773) had at least one infection recorded. The median number of parent-reported febrile episodes in the first year of life was 5 (2-10) and 82% of children (634/775) had at least one febrile episode reported. The median number of antibiotic prescriptions from birth to age 5 years was 1 (0-3) and 63% (583/928) received at least one antibiotic prescription. Of the 842, antibiotics were prescribed to 4%, 7%, and 15% of children in the 3, 6, and 12 months preceding vascular measurements, respectively. “JUST A SUGGESTION.

This implies to the second paragraph, mean and SD are always together. The author can simply say the mean (SD) of the cohort for CIMT was 388.6 um (42.3).

7. PLOS authors have the option to publish the peer review history of their article (what does this mean?). If published, this will include your full peer review and any attached files.

Reviewer #4: No

---

## [Author Response · Author response to Decision Letter 1]

11 Aug 2023

Reviewer #4:

Suggestion/s to the author:

The author should avoid repeating the numbers that are in tables (rather refer the reader to the specific table). If it is necessary, they can emphasize some of the key numbers for the reader that they think are most important. In the 1st paragraph of the results, I see that the author was showing the median and IQR, I suggest minor edits. Median and IQR are always together, therefore, no need to be continually repeating the "IQR". Maybe the author can write something like this “The median (IQR) of GP-diagnosed infections from birth to age 5 years was 3 (1-6) and 92% of children (711/773) had at least one infection recorded. The median number of parent-reported febrile episodes in the first year of life was 5 (2-10) and 82% of children (634/775) had at least one febrile episode reported. The median number of antibiotic prescriptions from birth to age 5 years was 1 (0-3) and 63% (583/928) received at least one antibiotic prescription. Of the 842, antibiotics were prescribed to 4%, 7%, and 15% of children in the 3, 6, and 12 months preceding vascular measurements, respectively. “JUST A SUGGESTION.

This implies to the second paragraph, mean and SD are always together. The author can simply say the mean (SD) of the cohort for CIMT was 388.6 um (42.3).

Response:

We are grateful to the reviewer for taking the time to make these detailed suggestions. We have edited the text as suggested, which makes the manuscript more concise and easier to read.

I have uploaded versions of the revised manuscript with tracked changes and a clean version.

---

## [Editor Report · Decision Letter 2]

13 Aug 2023

Childhood infection burden, recent antibiotic exposure and vascular phenotypes in preschool children

PONE-D-22-01018R2

Dear Dr. Burgner,

We’re pleased to inform you that your manuscript has been judged scientifically suitable for publication and will be formally accepted for publication once it meets all outstanding technical requirements.

Kind regards,

Engelbert Adamwaba Nonterah, MD, PhD

Academic Editor

PLOS ONE
---

## [Editor Report · Acceptance letter]

6 Sep 2023

PONE-D-22-01018R2

Childhood infection burden, recent antibiotic exposure and vascular phenotypes in preschool children

Dear Dr. Burgner:

I'm pleased to inform you that your manuscript has been deemed suitable for publication in PLOS ONE. Congratulations! Your manuscript is now with our production department.

Kind regards,

on behalf of

Dr. Engelbert Adamwaba Nonterah

Academic Editor

PLOS ONE